# SHORT REPORT

# The desmoplakin tail domain position in the desmosomal plaque is isoform dependent

Collin M. Ainslie[1],*, Krishna Patel[1],*, Yen T. B. Tran[1], Samuel C. Bartley[1], Navaneetha Krishnan Bharathan[2], Volker Spindler[3] and Alexa L. Mattheyses[1],‡

## ABSTRACT

Desmoplakin (DP, also known as DSP) is a key protein in desmosomes, cell–cell junctions that provide mechanical integrity to the skin and heart. DP has three isoforms, DPI, DPIa and DPII, which differ only in the length of their central rod domain and arise from alternative splicing. Alterations of tissue-specific DP isoform expression underlie rare skin and heart diseases. Desmosomes are macromolecular complexes, and their protein architecture is essential for physiological function. Here, we used direct stochastic optical reconstruction microscopy (dSTORM) to define the architectural arrangement of DPI, DPIa and DPII with a C-terminal mEGFP expressed in DP-knockout (KO) HaCaT cells. We show the DP tail domain position is isoform dependent and correlates with rod length. DPI has the longest rod domain, and its tail is farthest from the plasma membrane, whereas DPII has the shortest rod and is closest. This variable tail location architecture was conserved in wild-type HaCaT cells expressing both DPI and DPII. We propose a novel aligned angle model, with each DP isoform co-aligned at an acute angle relative to the plasma membrane. These results provide insight into how DP architecture supports desmosome function.

KEY WORDS: Desmosome, Desmoplakin, dSTORM, Architecture

## INTRODUCTION

Desmosomes are protein complexes that mediate cell–cell adhesion and provide integrity to tissues that experience mechanical stress (Kurn and Daly, 2024; Johnson et al., 2014; Bharathan et al., 2024). Dysregulation of desmosomes contributes to a variety of human diseases primarily impacting the skin and heart (Yuan et al., 2021; Mohammed and Chidgey, 2021; Stahley and Kowalczyk, 2015). Desmosomes are composed of the desmosomal cadherin family desmoglein (DSG) and desmocollin (DSC) proteins, the armadillo family plakophilin (PKP) and plakoglobin (PG, also known as JUP)

[1]Department of Cell, Developmental, and Integrative Biology, The University of Alabama at Birmingham, Birmingham, AL 35294, USA. [2]Departments of Dermatology and Cell and Biological Systems, Pennsylvania State University, College of Medicine, Hershey, PA 17033, USA. [3]Institute of Anatomy and Experimental Morphology, University Medical Center Hamburg -Eppendorf, Martinistrasse 52, 20246, Hamburg, Germany.
*These authors contributed equally to this work

‡Author for correspondence (mattheyses@uab.edu)

C.M.A., 0000-0002-2043-4223; K.P., 0009-0008-6033-8579; Y.T.B.T., 0009-0006-0468-9484; S.C.B., 0009-0007-1551-3318; N.K.B., 0000-0001-8340-5561; V.S., 0000-0002-1302-5421; A.L.M., 0000-0002-5119-7750

proteins, and desmoplakin (DP, also known as DSP) (Fig. 1A) (Kowalczyk et al., 1994; Kowalczyk and Green, 2013). Electron microscopy reveals desmosomes are organized into three distinct bands oriented parallel to the membrane and with mirror symmetry across the midline (Bharathan et al., 2024; Odland, 1958). The extracellular domain (ECD) is where DSC and DSG cadherin domains undergo trans binding. The DSG and DSC cytoplasmic tails, PG, PKPs and the head domain of DP localize to the outer dense plaque (ODP). The DP tail domain localizes to the inner dense plaque (IDP) (North et al., 1999; Stahley et al., 2016). The specific 3D arrangement of proteins within these domains remains unresolved.

DP is an obligate desmosomal protein responsible for anchoring desmosomes to intermediate filaments (IFs), such as keratins. DP has a tripartite structure with an N-terminal plakin head domain, coiled-coil central rod domain and IF-binding C-terminal tail domain (Bornslaeger et al., 2001; Kowalczyk et al., 1999; Choi et al., 2002). The DP rod domain forms homodimers, and mutations destabilizing dimerization impair desmosome integrity (Daday et al., 2019; Kowalczyk et al., 1994; Green et al., 1992). Alternative splicing of *DSP* generates three isoforms that are structurally identical except for rod domain length (Cabral et al., 2010; Green et al., 1988). DPI (332 kDa), DPIa (279 kDa) and DPII (260 kDa) have 888 amino acid (aa), 448 aa and 290 aa rod domains, respectively (Cabral et al., 2010; O'Keefe et al., 1989). The tissue expression of the isoforms is variable, with stratified epithelia expressing DPI and DPII at roughly equivalent levels, and cardiomyocytes primarily expressing DPI. DPIa is a minor isoform with substantially lower expression (Cabral et al., 2010). Essential roles of the isoforms are highlighted by rare human diseases. Loss of DPI expression caused by an isoform-specific homozygous nonsense variant leads to early onset cardiomyopathy (Uzumcu et al., 2006). Heterozygous premature stop codon variants generating either DPI or DPII haploinsufficiency can result in palmoplantar keratoderma (Armstrong et al., 1999; Whittock et al., 1999). Finally, knockdown of DPI or DPII has differential effects on desmosome cadherin composition and mechanical resistance in HaCaT keratinocytes (Cabral et al., 2012). These findings suggest expression of multiple DP isoforms in skin provide greater resistance to mechanical strain, and isoforms cannot substitute for one another in the heart. However, how DP isoforms individually contribute to desmosome function is unknown.

Desmosomes are complex, macromolecular assemblies essential in maintaining tissue integrity. Although the identity of desmosomal components is known, their architecture is not well defined. Desmosomal cadherins have a repeating antiparallel architecture in the ECD, but with a degree of flexibility (Dean and Mattheyses, 2022; Sikora et al., 2020). Electron tomography identified a repeating pattern of protein densities in the ODP, relating to PKP, PG and the DP head domain (Al-Amoudi et al., 2011). Unlike other desmosomal components, DP spans the ODP and IDP. DP–keratin binding has

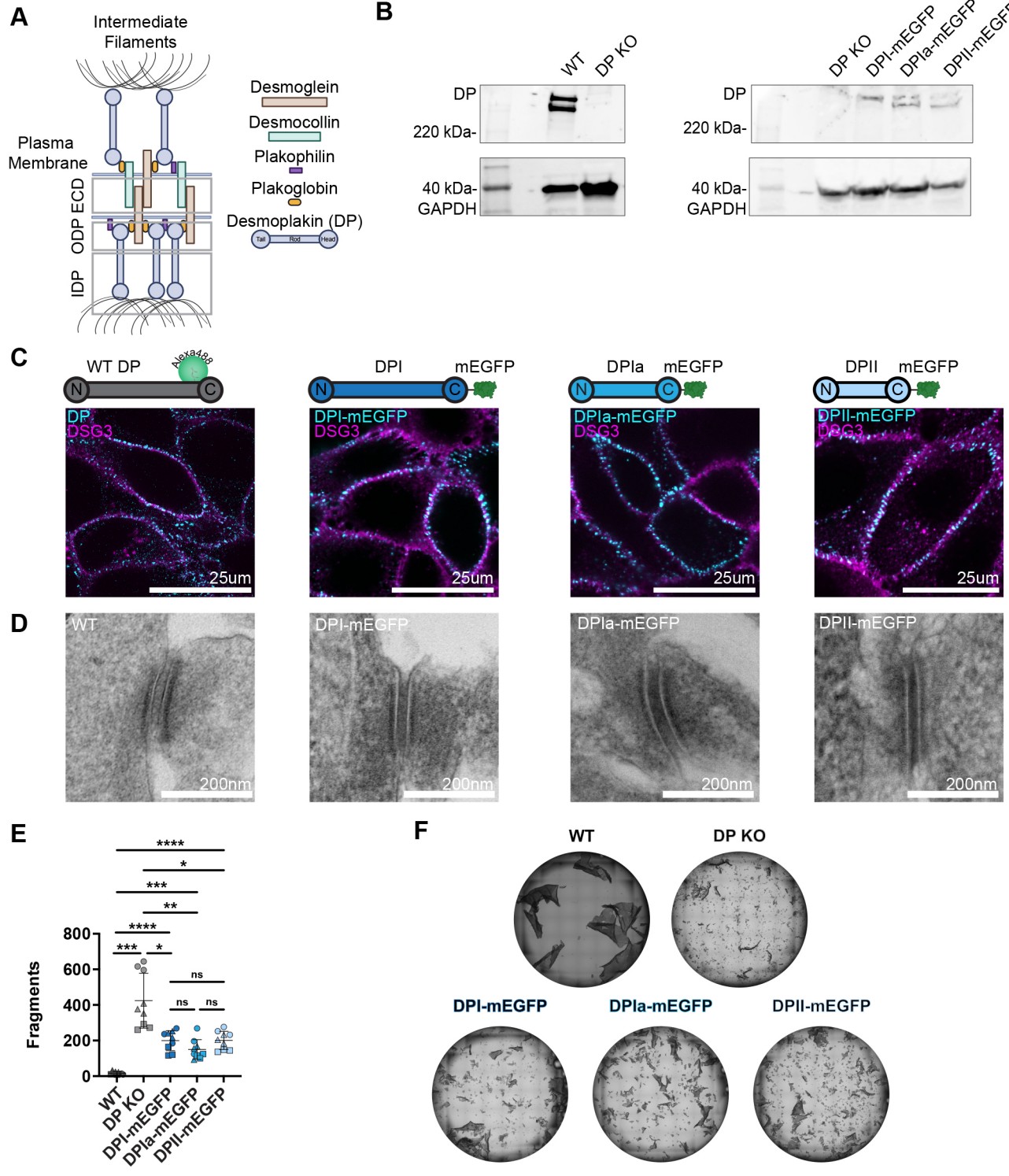

**Fig. 1. Characterization of HaCaT cells expressing desmoplakin isoforms.** (A) Desmosome schematic. (B) Representative western blots showing DP expression in WT and DP KO HaCaT cells (left; three independent replicates) and DP–mEGFP isoform expression in the DPI–, DPIa– and DPII–mEGFP stable cells (right; four independent replicates). The predicted molecular masses are: DPI, 332 kDa; DPII, 260 kDa; DPI–mEGFP, 359 kDa; DPIa–mEGFP, 306 kDa; DPII, 287 kDa; GAPDH, 37 kDa (see Fig. S2). (C) Representative maximum projection confocal images of DP (cyan) and DSG3 (magenta) in WT HaCaT cells and DP–mEGFP stable cells. Images are representative of at least three independent replicates. Scale bars: 25 μm. (D) Representative TEM images of desmosomes in WT and DP–mEGFP isoform-expressing HaCaT cells. Images are representative of at least five desmosomes for one independent replicate. Scale bars: 200 nm. (E) Number of fragments from a dispase cell adhesion assay for WT, DP KO and DP–mEGFP isoform HaCaT cells (mean±s.d., n=9 with three cell sheets in each of three biological replicates, each replicate with a unique symbol shape). WT HaCaT, 16±8; DP KO HaCaT, 424±154; DPI–mEGFP, 200±56; DPIa–mEGFP, 149±56; DPII–mEGFP, 201±51 (mean±s.d.). Shapiro–Wilk test confirmed normal distribution (P>0.05). Brown–Forsythe test determined the variance was not equal between groups (P<0.05). Brown-Forsythe and Welch ANOVA with Dunnett's t3 multiple comparison test assessed the statistically significant differences (****P<0.0001; ***P<0.001; **P<0.01; *P<0.05; ns, not significant). (F) Representative images from the dispase cell adhesion assay in E. Images show a 22 mm diameter.

been defined, but the architecture of this interaction within the IDP has yet to be characterized (Kroger et al., 2013). DP head and tail domain plaque positions have been mapped through immunogold EM and direct stochastic optical reconstruction microscopy (dSTORM) (North et al., 1999; Stahley et al., 2016). Interestingly, this imaging data combined with *in vitro* measurement of DP head to tail 'length' suggest that the long axis of DP is not oriented perpendicularly to the plasma membrane, but instead at an acute angle relative to the membrane (Stahley et al., 2016; O'Keefe et al., 1989). Previously, we found that DP tail domain localization correlates with function during desmosome maturation and with ODP protein composition (Stahley et al., 2016; Beggs et al., 2022). However, none of these studies compared DP isoforms and, in many cases, multiple isoforms were indistinguishable. Given the importance of DP isoform expression to desmosome function, deciphering their architectural arrangement is essential to understand the junction's role in combating mechanical strain. In this study, we use human keratinocyte HaCaT cells expressing single DP isoforms and super-resolution dSTORM to uncover isoform dependent architecture.

## RESULTS
### Characterization of HaCaT cells expressing DP-mEGFP isoforms

DP rod domain length varies between isoforms, but whether and how this affects desmosome structure and function is not understood. We sought to determine how the isoform-specific rod lengths impact DP architecture. Two models capture distinct possible DP architectures: the 'uniform tail position' model hypothesizes the tail domain of each isoform is the same distance away from the plasma membrane, creating a single interface for IF binding. In contrast, the 'variable tail location' model hypothesizes the tail domain position is dependent on the rod domain length of each isoform. In both models, the DP head domain location is consistent across all isoforms.

A central challenge to studying DP isoform-specific architecture is sequence identity. Specific antibodies for the smaller isoforms, DPIa and DPII, are not available, preventing their individual labeling in wild-type (WT) cells, which express multiple isoforms. To address this challenge, we stably expressed DPI, DPIa or DPII with a C-terminal mEGFP tag in CRISPR/Cas9 engineered DP-knockout (KO) HaCaT human keratinocytes generating three HaCaT cell lines each expressing one DP–mEGFP isoform (Wanuske et al., 2021). Although DP was not detected in the KO HaCaTs, overexpression showed each DP–mEGFP isoform (Fig. 1B). All DP–mEGFP constructs colocalized with DSG3 at cell borders, similar to what is seen in WT HaCaT cells (Fig. 1C). Although desmosomes are not present in the parental KO HaCaTs, we found that desmosome ultrastructure was indistinguishable between WT- and the DP–mEGFP isoform-expressing cells (Fig. 1D). Expression of DP–mEGFP isoforms improved resistance to mechanical strain, which is severely disrupted by DP KO (Fig. 1E,F). Interestingly, although there was no significant difference in fragmentation between isoforms, indicating a similar capacity to resist mechanical strain, cells expressing a single isoform resisted mechanical strain less effectively than WT HaCaTs. This difference suggests the presence of multiple DP isoforms might increase desmosome adhesion strength. Thus, DP–mEGFP isoform expression recapitulates desmosome formation, localization and ultrastructure in DP KO HaCaTs.

### DP isoform molecular maps

Next, we wanted to determine the architectural arrangement of the DP isoforms. To do so, we conducted super-resolution dSTORM on the DP–mEGFP-expressing cells labeled with anti-GFP nanobodies

conjugated to Alexa Fluor 647. In all cell lines, DP appeared as puncta in diffraction-limited widefield microscopy. dSTORM revealed two individual plaques within these puncta, each belonging to one half of a desmosomal junction (Fig. 2A; Stahley et al., 2016). To extract architectural information from these images, we quantified the distance between the plaques for many individual desmosomes in each group. Desmosomes have mirror symmetry across the midline and this 'plaque-to-plaque' distance represents the average separation of mEGFP on the tail domain of DP across neighboring cells (Fig. S1) (Stahley et al., 2016; Odland, 1958). The larger the plaque-to-plaque distance, the farther away from the membrane the tagged domain is located. We found the plaque-to-plaque distances for DPI–mEGFP, DPIa–mEGFP and DPII–mEGFP tail domains were 192±26 nm, 143±18 nm and 118 ±14 nm, respectively (mean±s.d.; Fig. 2B,C). These data indicate the position of the DP tail domain in the plaque is isoform dependent and the distances of the DP tail domain from the midline correlates with the varying length of DP isoform rod domains.

Next, we tested whether the differences in tail domain location could be explained by a change in the overall position of DP. To do this, we labeled the DP–mEGFP cells with an antibody against the DP head domain and conducted dSTORM. The DP head domain plaque-to-plaque distance was smaller than the tail domain, reflecting its membrane proximal position. Plaque-to-plaque distances were similar for all isoforms (DPI–mEGFP, 70±12 nm; DPIa–mEGFP, 60±13 nm; DPII–mEGFP, 65±15 nm) (Fig. 2D,E). The relatively minor 10 nm differences, while statistically significant, are not sufficient to explain the 68 nm difference in the tail domain plaque-to-plaque distance. These findings suggest the variable localization of DP isoform tail domains is not a result of head domain localization.

Finally, we asked whether co-expression of multiple DP isoforms influences DP architecture. DP KO HaCaT cells were transfected with both DPI–mEGFP and DPII–mCherry, where both tags are on the C-terminus of the protein. We observed the DPI tail domain extends farther into the cytosol than that of DPII (Fig. 3A,B). This arrangement was conserved across many individual desmosomes (Fig. 3C). To next quantify the nanoscale architecture of DP in cells expressing both DPI and DPII, we transfected DPI–mEGFP or DPII–mEGFP into WT HaCaT cells. Following staining with an anti-GFP nanobody, we used dSTORM to quantify the tail domain architecture of DPI and DPII in the WT background. We found the plaque-to-plaque distance of DPI–mEGFP and DPII–mEGFP tail domains to be 211±34 nm and 152±23 nm, respectively (Fig. 3D,E). This data provides additional evidence supporting that the variable location of the DP isoform tail domain is rod domain length dependent.

Finally, we determined a possible arrangement of DP within the plaque using a simple model defined by the angle between the long axis of DP and the plane of the plasma membrane, which we call the angle of alignment ($\theta_{DPX}$). Published data from rotary shadow electron microscopy determined the average length of DPI to be 162 nm, with the head and tail domain determined to be 16 nm each (O'Keefe et al., 1989). From this, we determined that the DPI rod length is 130 nm. With the assumptions that rod length correlates with amino acid number and the head and tail domain sizes are conserved between isoforms, we calculated the length of DPIa and DPII to be 97 nm and 75 nm, respectively. The calculated length of DPII is within the range reported by O'Keefe (DPIa was not included in that study), supporting the validity of our estimation (O'Keefe et al., 1989). Using DP length and empirically measured head and tail domain positions from the DP–mEGFP isoform stable cells, we determined $\theta_{DPX}$, to be 22°±6°, 25°±8° and 21°±9°

Journal of Cell Science

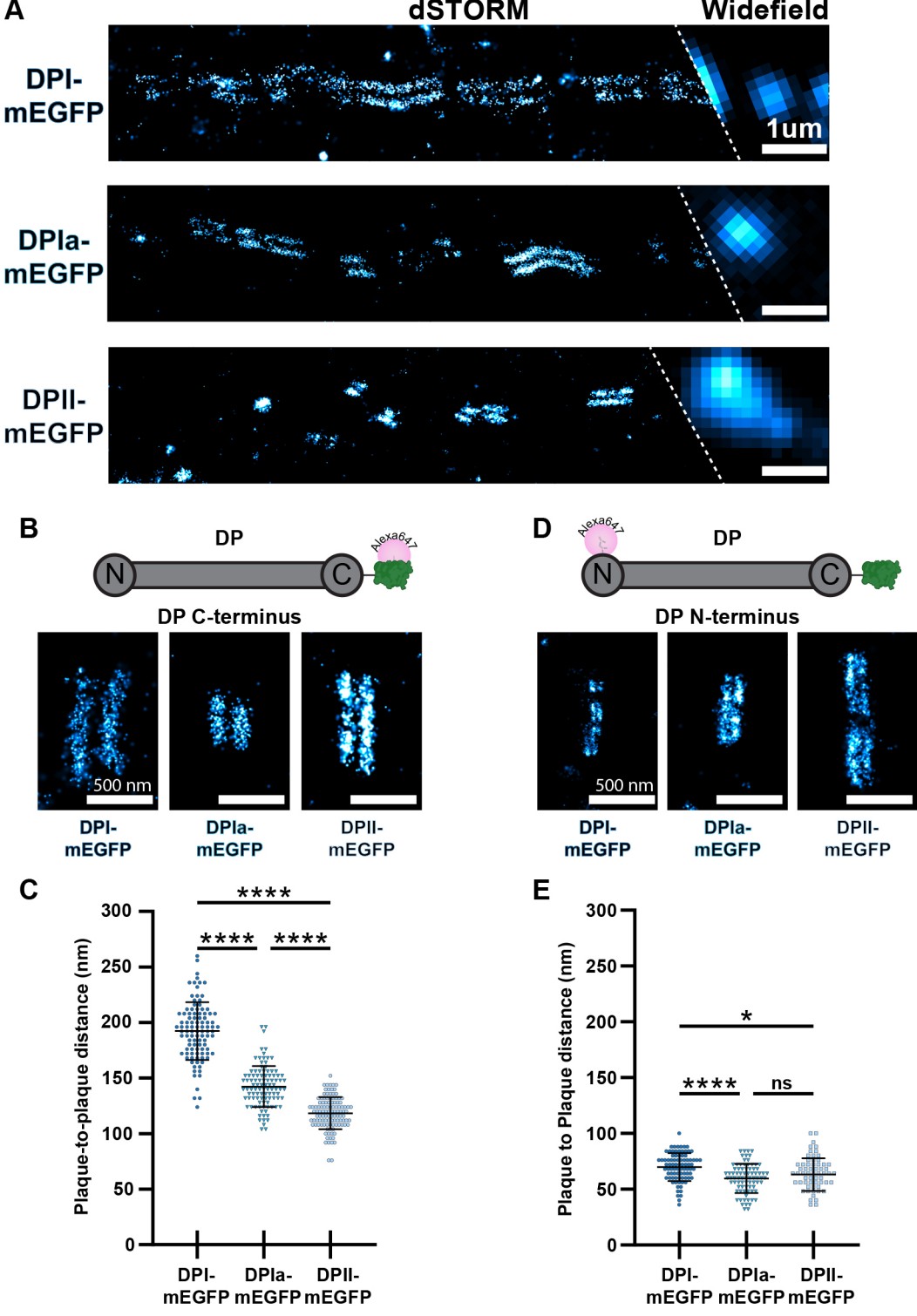

**Fig. 2.** See next page for legend.

(mean±s.d.) for DPI, DPIa and DPII, respectively (Fig. 4A). The consistency in $\theta_{DPX}$ strongly support the aligned angle model to describe the isoform dependent variable tail locations.

## DISCUSSION

The work presented here establishes a model for the architectural arrangement of the three known DP isoforms. The tail domain

position of DPI, the longest isoform, is the farthest from the midline and that of DPII, the shortest isoform, is the closest to the midline. DPIa is intermediate in length, and correspondingly its tail domain is located between that of DPI and DPII. This variable tail localization of the DP isoforms generates exciting questions about the integration of keratin into the complex. The DP–keratin interaction is essential to both desmosome and keratin function.

**Fig. 2. dSTORM reveals the architecture of desmoplakin isoforms.**
(A) Representative dSTORM and widefield images of DP–mEGFP isoform
HaCaT cells labeled with anti-GFP nanobody. Scale bars: 1 µm. (B)
Representative dSTORM images of single desmosomes from DP–mEGFP
isoform HaCaT cells labeled with Alexa Fluor 647 anti-GFP nanobody
representing the DP tail domain. Scale bars: 500 nm. (C) Scatter plot of the
tail domain plaque-to-plaque distance for each isoform, each data point
represents an individual desmosome. DPI–mEGFP, $n$=101, 192±26 nm;
DPIa–mEGFP, $n$=101, 143±18 nm; DPII–mEGFP, $n$=112, 118±14 nm ($n$ is
number of desmosomes, mean±s.d.). (D) Representative dSTORM images
of single desmosomes from DP–mEGFP isoform HaCaT cells labeled with a
primary antibody for the DP head domain and an Alexa Fluor 647 secondary
antibody representing the DP head domain. Scale bars: 500 nm. (E) Scatter
plot of head domain plaque-to-plaque distances for each isoform. DPI–
mEGFP, $n$=84, 70±12 nm; DPIa–mEGFP, $n$=79, 60±13 nm; DPII–mEGFP:
$n$=60, 65±15 nm (number of desmosomes, mean±s.d). In C and E,
desmosomes were analyzed from a minimum of three independent
replicates. Shapiro–Wilks test for normality confirmed data was normally
distributed ($P$>0.05 for each isoform). Brown–Forsythe test determined the
variance to not be equal between groups ($P$<0.05). Brown–Forsythe and
Welch ANOVA with Dunnett's t3 multiple comparison test assessed the
statistically significant differences of DP tail domain position. Statistically
significant differences of the DP head domain position were assessed by
one-way ANOVA with Tukey's multiple comparison test (**** $P$<0.0001,
*$P$<0.05, ns not significant). Error bars are mean±s.d.

Post-translational modifications to the keratin-binding domain of
DP directly influence desmosomal strength (Bartle et al., 2020;
Hobbs and Green, 2012; Perl et al., 2023). Additionally, changes to
keratin isoform expression impact DP localization and desmosome
strength (Wang et al., 2018). Although the DP–keratin interface has
been examined, how it varies between isoforms has yet to be
described. The isoform-dependent tail domain position suggests
that keratin could bind to DP at multiple layers relative to the plasma
membrane. One scenario could involve one keratin bundle bound
simultaneously by multiple DP isoforms at different positions as it
loops though the IDP (Fig. 4B). In this variable tail location
architecture, multiple keratin–DP interaction interfaces are present,
located either distal from the membrane for DPI or proximal to it for
DPII (Fig. 4C). Additionally, the dispase fragmentation assay raises
interesting questions regarding DP isoform dependence in the
resistance to mechanical strain. Although the data presented here
suggest there is no significant difference between isoforms,
differences in adhesive function between DPI and DPII have been
demonstrated by Cabral et al. (2012) using a unique siRNA
approach. Additional work is needed to fully understand how and
why desmosomes provide mechanical strength to cells. In skin
epithelia, where DPI and DPII are expressed at relatively equal
levels, such an interface could provide an advantage to resisting
mechanical strain. It will be fascinating to study whether greater
functional integrity is supplied by multiple linkages and how
variable DP rod domain lengths contribute to function. It is
important to note that all measurements here were made using
fluorescently tagged DP constructs. DP with a C-terminal GFP tag
has been used to study cadherin clustering, keratin filament
assembly, the dynamics of desmosome formation and fusion, and
can promote hyper-adhesion (Hobbs and Green, 2012; Bartle et al.,
2020; Broussard et al., 2017; Godsel et al., 2005; Bharathan et al.,
2023; Moch et al., 2020; Wanuske et al., 2021). Here we ascertained
the localization, desmosome ultrastructure and ability to restore
resistance to mechanical stress of the DP–mEGFP constructs in
DP KO HaCaTs (Fig. 1C–F). Together, this supports the ability
of C-terminal tagged DP to functionally recapitulate WT DP.
Additionally, desmosome architecture is broadly conserved across
cells and tissues. The nanoscale DP–mEGFP architecture described

here is highly similar to that shown in our previous work examining
endogenous DP in A431, HaCaT, MDCK and HUC cells and
primary human keratinocytes (Stahley et al., 2016; Beggs et al.,
2022). Defining the architecture of untagged DP isoforms remains
an important future step, confounded by the challenge of labeling
the isoforms independently.

Data from our laboratory and others have suggested that DP
cannot be fully extended in the desmosomal plaque. Several
alternative DP arrangements have been proposed, including an
accordion fold or a 90° bend in the DPI rod (North et al., 1999;
Garrod and Chidgey, 2008; Stokes, 2007). These models provide
numerous lines of investigation into DP architecture; however, they
assume a uniform tail location across isoforms, in conflict with the
variable tail location presented here. To most simply explain our
data, we propose the DP isoforms are oriented at a similar angle in
the plaque. This is based on the assumption the rod domain length
scales linearly with number of amino acids. Different DP
architectures with folded or bent rod domains, possibly also
combined with an angle, could also fit the variable tail location data.
We note that the mean plaque-to-plaque distance for DPI and DPII
tail was different between the WT and DP KO HaCaT background.
This could be a biological difference, where the isoforms influence
one another. Alternatively, it could result from minor differences in
the cell confluency which impacts desmosome maturity, both of
which have previously been shown to impact DP tail domain
location (Stahley et al., 2016; Beggs et al., 2022). Together, this
work suggests a range of DP tail domain locations based on
desmosome assembly or maturity, supporting an analog mechanism
of tuning which would arise from a variable $\theta_{DPX}$.

One caveat of the aligned angle model is that it does not account
for variations in structural flexibility or conformational changes in
the DP rod domain. One source of uncertainty in the model is the
sparsity of structural information regarding the DP rod domain.
Previous studies characterized the DP rod as a 'coiled coil' that
forms homodimers, and variants that impede homodimer formation
are associated with disease (Green et al., 1990; Mohammed and
Chidgey, 2021). Additionally, DP is mechanically sensitive;
although it is not under tension in steady state conditions,
exposure to mechanical strain induces tension across DP (Price
et al., 2018; Sadhanasatish et al., 2023). Our experiments were
conducted under steady-state conditions, with no external forces.
Understanding how DP isoforms respond to mechanical strain
will provide additional insights into the architecture and flexibility
of the rod domain. High-resolution structures and biophysical
studies of DP are crucial to defining the molecular architecture of
desmosomes.

Overall, this work provides novel insights into the collaborative
role of DP isoforms in desmosome architecture. This is one of the first
studies, to our knowledge, to examine the architecture of the DP
isoforms. We found a variable tail localization of DP isoforms and
propose an aligned angle model where all DP isoforms are oriented
with their long axis at a similar θ. Our discovery of the variable tail
location of DP isoforms will lead to new hypotheses about the
structure and function of the DP–IF interface. This model will be of
importance for example in the understanding of differences between
cardiac and epithelial desmosomes, which express one or two DP
isoforms. The co-expression of DPI and DPII, presenting two
different keratin-binding interfaces, could provide redundancy,
improving the ability of skin epithelia to resist diverse mechanical
forces. This could be in the form of multiple layers of keratin
integration or keratin looping between the isoforms, having multiple
anchoring points. In contrast, cardiomyocytes, with more uniform

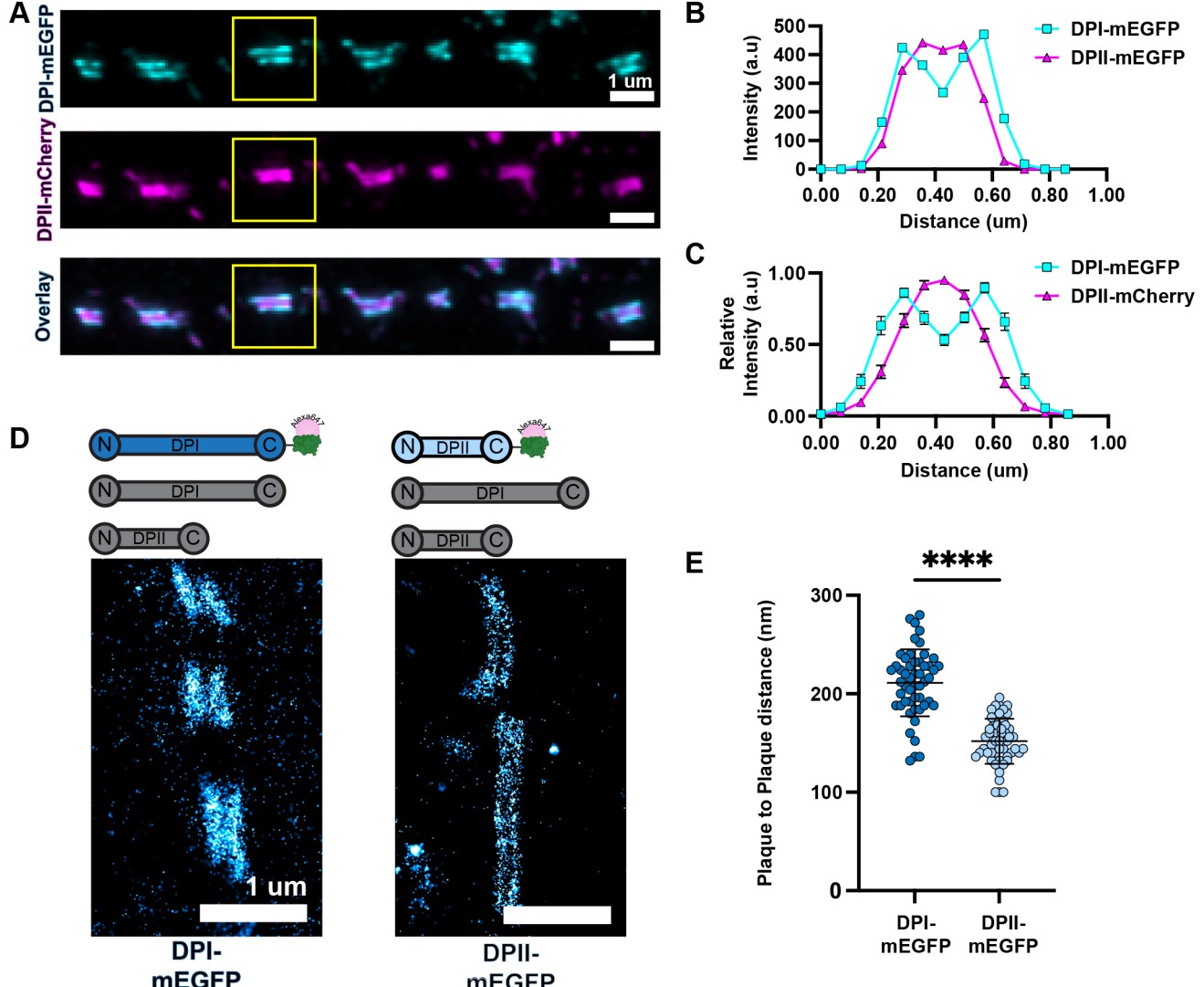

**Fig. 3. DP tail architecture is isoform dependent in cells expressing both DPI and DPII.** (A) Representative NSPARC images of DPI–mEGFP (cyan) and DPII–mCherry (magenta) in HaCaT DP KO cells. Scale bars: 1 µm. (B) Intensity line scan through the boxed desmosome in A. (C) Average relative intensity line scan of 63 desmosomes expressing DPI–mEGFP and DPII–mCherry. a.u., arbitrary units. Error bars indicate 95% confidence interval. (D) Representative dSTORM images of WT HaCaTs expressing DPI–mEGFP or DPII–mEGFP labeled with Alexa Fluor 647 anti-GFP nanobody to image the tail domain. Scale bars: 1 µm. (E) Scatter plot of DP tail domain plaque-to-plaque distance of DPI–mEGFP or DPII–mEGFP in WT HaCaTs. DPI–mEGFP, n=51, 211±34 nm; DPII–mEGFP, n=53, 152±23 nm (mean±s.d). n is the number of desmosomes which were imaged from at least two independent experiments. Shapiro–Wilks test for normality confirmed data was normally distributed (P>0.05 for each isoform). Unpaired two-tailed t-test revealed there is a significant difference in the tail domain plaque-to-plaque distance between DPI and DPII-mEGFP in WT HaCaTs (****P<0.0001). Error bars are mean±s.d.

mechanical forces, might have only one DP–IF anchoring interface (but with more complex junctional identities). This work highlights how a greater understanding of DP isoform structure and architecture and the DP–keratin interface is crucial for a full understanding of desmosome structure and function in health and disease.

## MATERIALS AND METHODS
### Cell culture
HaCaT cells were maintained in Dulbecco's modified Eagle's medium (DMEM; Corning, Tewksbury, MA, USA) supplemented with 10% fetal bovine serum (A5670801, Gibco) and 2% penicillin-streptomycin (15070063, Gibco) at 37°C and 5% $CO_2$. DP KO HaCaT cells were previously generated (Wanuske et al., 2021).

DP–mEGFP constructs were cloned to insert mEGFP following a linker (DPPVAT) on the C-terminus of each DP isoform in a pRP[Exp]-CMV mammalian gene expression vector from Vector Builder (Chicago,

IL, USA). DPI–mEGFP, DPIa–mEGFP or DPII–mEGFP was transfected into DP KO HaCaT cells using Lipofectamine 3000 following the manufacturer instructions (LC3000015; Thermo Fisher Scientific, Waltham, MA, USA). Two days following transfection the GFP-expressing population was enriched using fluorescence activated cell sorting (FACS) (BD FACSMelody; BD BioSciences, Franklin Lakes, NJ, USA). DP–mEGFP cells were maintained in medium as described above supplemented with 500 µg/ml of geneticin (10131027, Gibco).

Co-expression of DP isoforms was achieved through transfecting DPI or DPII–mEGFP into WT HaCaTs or DPII–mCherry into DPI–mEGFP DP KO HaCaT stable cells using Lipofectamine 3000 following the manufacturer's instructions (LC3000015, Thermo Fisher Scientific). Two days following transfection, samples were fixed with prepared for immunofluorescence.

### Antibodies
Antibodies used for immunofluorescence were: FluoTag-X4 anti-GFP conjugated with Alexa Fluor 647 (used for dSTORM, 1:250, N0304

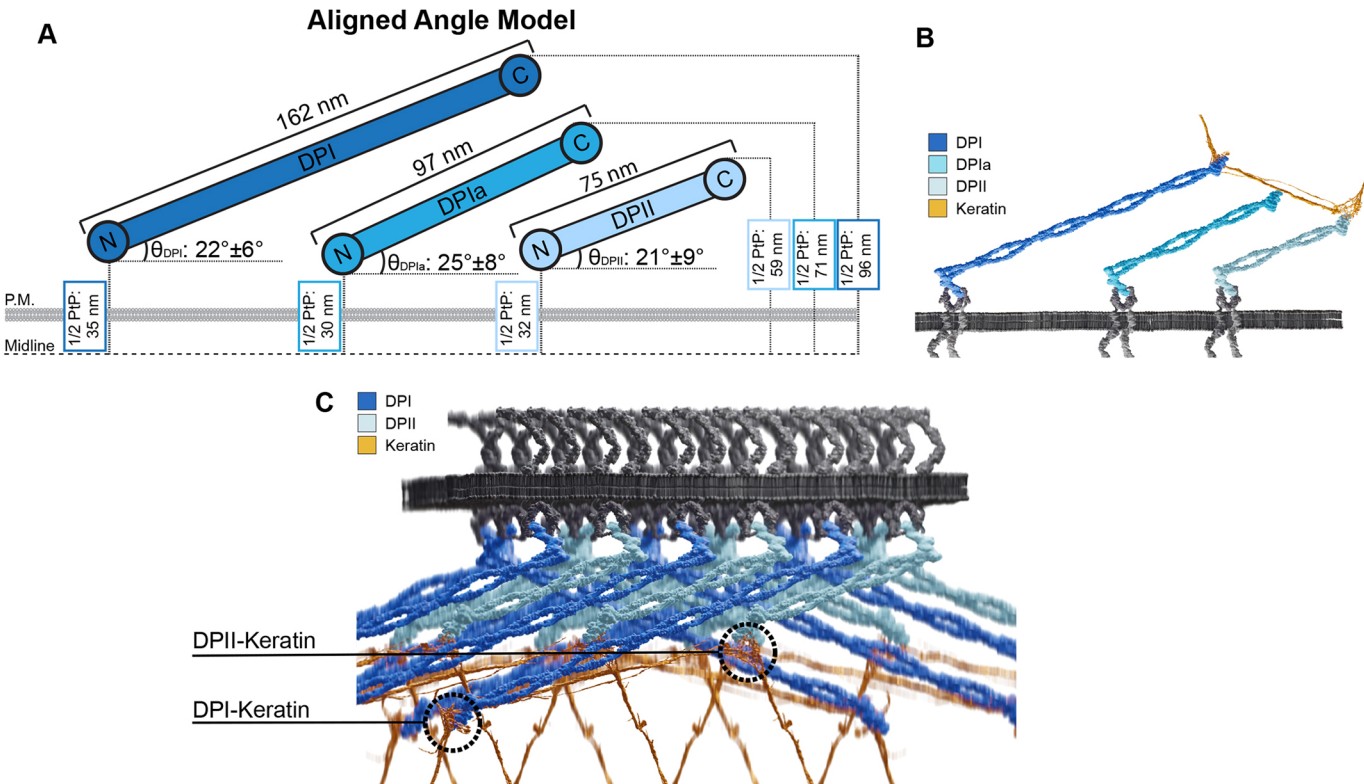

**Fig. 4. The aligned angle model describes the variable tail domain location of the DP isoforms.** (A) To scale schematic of DPI, DPIa and DPII architecture relative to the desmosome midline. Mean half plaque-to-plaque (1/2 PtP) distances were measured through our dSTORM analysis of each isoform. DP isoform lengths and angles of alignment were calculated as described in the text. There is cohesion in the predicted angle between the plane of the plasma membrane and the long axis of desmoplakin across the isoforms with $\theta_{DPI}$: 22°±6°; $\theta_{DPIa}$, 25±8°; and $\theta_{DPII}$, 21°±9° (mean±s.d.). (B) Illustration of the aligned angle model with one keratin filament bundle binding to the tail domains of all three DP isoforms. (C) 3D model of a desmosomal plaque illustrating DPI and DPII isoforms with keratin looping through the plaque. Note the positions of DPI- and DPII-keratin interactions relative to the plasma membrane. Cadherins are arranged following anti-parallel model. Illustrations in B and C were created using Blender 4.0.

NanoTag Biotechnologies, Gottingen, Germany); FluoTag-X4 anti-GFP conjugated to Atto488 (1:250, N0304 NanoTag Biotechnologies); anti-desmoplakin (1:150, A303-356A Bethyl Laboratories, Montgomery, TX, USA); anti-desmoplakin (N-terminal/head domain antibody used for dSTORM; 1:100, 00192 BiCell Scientific, Maryland Heights, MO, USA); anti-desmoglein 3 (DSG3; 1:500, 20483 Cell Signaling, Danvers, MA, USA); anti-rabbit Alexa Fluor 647 (1:1000, used for dSTORM, a-21244 Thermo Fisher Scientific, Waltham, MA, USA); anti-rabbit-IgG Alexa Fluor 488 (1:1000, A-11008 Thermo Fisher Scientific Waltham, MA, USA). Antibodies for western blotting were anti-desmoplakin [1:2000, EPR4383(2) Abcam, Waltham, MA, USA]; anti-GAPDH (1:2000, 6C5 Cell Signaling, Danvers, MA, USA); anti-mouse IgG HRP-linked (1:1000, 7076 Cell Signaling, Danvers, MA, USA); and anti-rabbit IgG HRP-linked (1:1000, 7074 Cell Signaling, Danvers, MA, USA).

### Immunofluorescence

Cells were grown on coverslips coated with fibronectin. Between 1 and 2 days of confluency, samples were washed 1× with PBS, fixed and washed 3× (5 m) with PBS [0.5% Triton X-100, 0.5% normal goat serum (NGS; 16210-064, Gibco) and 0.5% normal horse serum (NHS; 16050-130, Gibco)]. Samples were blocked for 30 m at room temperature in blocking buffer [PBS with 0.5% Triton X-100, 5% NGS, 5% NHS and 10 mg/ml bovine serum albumin (BSA)]. Following blocking, samples were incubated at 37°C in a humidity chamber (1 h and shaking at 35 rpm) with primary antibody. Samples requiring a secondary antibody were then washed 3× (5 m) with PBS (with 0.5% Triton X-100, 0.5% NGS and 0.5% NHS) and incubated for 30 m at 37°C with secondary antibody a concentration of 1:1000. Samples were then washed 3× (5 min) with PBS and stored at 4°C protected from light until imaging.

### Microscopy

Confocal and Nikon Spatial Array Confocal (NSPARC) images were obtained on a Ti-2 AXR microscope (Nikon Instruments, Melville, NY, USA) with a 60×1.42 NA oil immersion objective, and 488 and 647 nm laser excitation with Nyquist sampling. Z-stacks were acquired and deconvolved with the Richardson–Lucy algorithm with eight iterations for confocal and 15 iterations for NSPARC.

dSTORM images were obtained on a Nikon Ti-2 microscope system equipped with a 100×1.49 NA oil immersion objective, 647 nm laser and Andor iXon EMCCD camera (Oxford Instruments, Abingdon, UK). 10,000 frames were acquired for each image. Samples were imaged in a 50 mM Tris-HCl pH 8.0, 10 mM NaCl, 10% glucose buffer with 5% 1 M MEA (Sigma, St. Louis, MO, USA) and 2% GLOX [20% 17 mg/ml catalase (Roche, Penzberg, Germany) and 14 mg glucose oxidase (Sigma, St. Louis, MO, USA)] each prepared in 50 mM Tris-HCl and 10 mM NaCl.

Transmission electron microscopy (TEM) sample preparation was as described in Bartle et al. (2020). TEM was performed on a JOEL 1400 HC Flash TEM (Jeol USA, Peabody, MA, USA) at 120 kV with an AMT NanoSprint43 Mk-II camera (AMT, Woburn, MA, USA).

### Image analysis

dSTORM images were exported at 4 nm/pixel and analyzed using our in-house MATLAB (Mathworks, Natic, MA) analysis pipelines as described in Stahley et al. (2016) and Beggs et al. (2022). Desmosomes were manually selected and then automatically excised and aligned. Intensity was measured across the junction and averaged along the desmosome length. Linescans were normalized, smoothed and the plaque-to-plaque distance was quantified with the 'peakfinder' function. See Fig. S1C for analysis details.

## Western blotting

Cells were grown to confluency, washed with ice-cold PBS supplemented with a protease inhibitor cocktail (Roche Diagnostics GmbH, Germany) and scraped on ice. Samples were collected and centrifuged at 828 $g$ at 4°C (5 m). The supernatant was aspirated, pellet was resuspended and lysed in 8 M urea on ice (30 m). Samples were centrifuged at 13,523 $g$ at 4°C (15 m). The supernatant was collected and stored at −80°C. A BCA was utilized to determine protein concentration and either 85 or 15 μg of protein was loaded in 4–15% gradient gels for gel electrophoresis. 15 μg was utilized to confirm no DP was detectable in DP KO HaCaT cells, and 85 μg was used to confirm expression of DP–mEGFP constructs. The samples were transferred to a PVDF membrane overnight at 4°C. The PVDF membranes incubated in Intercept Blocking Buffer (LiCor, Lincoln, NE, USA) at room temperature (RT; 1 h), incubated in primary antibody diluted in Intercept Antibody Diluent (LiCor) overnight at 4°C, washed 3× with PBS with 0.1% Tween 20 (PBST) and incubated in secondary antibody diluted int Intercept Antibody Diluent at RT (1 h), washed 3× with PBST, and imaged and analyzed on Bio-Rad ChemiDoc MP imaging system (Bio-Rad, Hercules, CA, USA). Full blots and replicates are in Fig. S2.

## Dispase fragmentation assay

Cells were grown to confluency and treated with 1 U/ml dispase (Sigma, St. Louis, MO, USA) until the cell sheets were lifted. Using an FBS-coated 1000 μl pipette tip, cell sheets were fragmented with an Eppendorf Xplorer automated pipette with the number of aspirations and force uniform across all samples (Eppendorf, Hamberg, Germany). Following fragmentation, samples were fixed with 1% paraformaldehyde (Electron Microscopy Sciences, Hatsfield, PA, USA) and stained with 0.2% Methylene Blue (Ricca Chemical Company, Arlington, TX, USA) overnight. Fragments were counted and imaged. Nine biological replicates were acquired from three independent experiments.

## Statistical analysis

Shapiro–Wilks test was used to test for normality and the Brown–Forsythe test to test for equal variance of samples when three or more samples were assessed. If the Shapiro–Wilks and Brown–Forsythe tests requirements were met, one-way ANOVA with Tukey's multiple comparison tests were used. When the requirements of the Brown-Forsythe test were not met, a Brown–Forsythe and Welch ANOVA with Dunnett's t3 multiple comparison test was used. For two-sample comparisons, the Shapiro–Wilks test to test for normality was used followed by an unpaired two-tailed $t$-test. GraphPad Prism was used for all statistical analyses and graph generation (Version 8.3.1).

## Resources

All resources and software are available upon request by contacting the lab.

## Acknowledgements

The authors thank the High-Resolution Imaging Facility at the University of Alabama in Birmingham for the excellent support and assistance with dSTORM and TEM. The UAB High Resolution Imaging Facility was supported by NCI P30 CA013148.

## Competing interests

The authors declare no competing or financial interests.

## Author contributions

Conceptualization: C.M.A., K.P., A.L.M.; Formal analysis: C.M.A., K.P.; Funding acquisition: A.L.M.; Investigation: C.M.A., K.P., Y.T.B.T., S.C.B.; Methodology: C.M.A., K.P., V.S., A.L.M.; Resources: V.S.; Supervision: A.L.M.; Validation: C.M.A., K.P.; Visualization: C.M.A., K.P., N.K.B., A.L.M.; Writing – original draft: C.M.A., K.P.; Writing – review & editing: C.M.A., K.P., Y.T.B.T., N.K.B., S.C.B., V.S., A.L.M.

## Funding

This work was supported by the National Institutes of Health (NIH) under grant R01 AR072697 to A.L.M. Open Access funding provided by University of Alabama at Birmingham. Deposited in PMC for immediate release.

## Data and resource availability

All relevant data is found within the article and supplementary material.

## First Person

This article has an associated First Person interview with Collin Ainslie and Krishna Patel, co-first authors of the paper.

## Peer review history

The peer review history is available online at https://journals.biologists.com/jcs/lookup/doi/10.1242/jcs.263906.reviewer-comments.pdf

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
