## [Peer Review File · Journal of Cell Science]

Desmoplakin tail domain position in the desmosomal plaque is isoform dependent

Collin M. Ainslie, Krishna Patel, Yen T. B. Tran, Navaneetha Krishnan Bharathan, Samuel C. Bartley, Volker Spindler and Alexa L. Mattheyses
DOI: 10.1242/jcs.263906

Editor: Kathleen Green

Review timeline

Original submission:	5 February 2025
Editorial decision:	4 March 2025
First revision received:	5 June 2025
Editorial decision:	1 July 2025
Second revision received:	29 July 2025
Accepted:	30 July 2025

Original submission

First decision letter

MS ID#: jcs.263906

MS TITLE: Desmoplakin tail domain position in the desmosomal plaque is isoform dependent

AUTHORS: Collin M Ainslie; Krishna Patel; Yen T. B. Tran; Navaneetha Krishnan Bharathan; Volker Spindler; Alexa L Mattheyses

ARTICLE TYPE: Short Report

Dear Alexa,

We have now reached a decision on the above manuscript.

To see the reviewers' reports and a copy of this decision letter, please go to:

Reviewer 1

Advance summary and potential significance to field

The report provides evidence for different localization of Desmoplakin isoforms based on dSTORM superresolution microscopy. This may explain why the different isoforms interact differently at the plasma membrane interface of desmosomes.

The study is very interesting conceptually, and the imaging is of very high resolution and well-controlled. There are, however, some points that currently limit the potential impact of the finding.

Comments for the author

In various cell types the desmoplakin isoforms might be co-expressed in cells. Can the authors test whether the intrinsic difference of the isoforms in plaque to plaque differences are maintained if DPI, DPIa and/or DPII are co-expressed? This would answer the question whether the organization of one Dp isoform affects another and will test the provided model in figure 3.

The expression of the Dp isoforms induces desmosome formation in these KO HaCats, however, one wonders what results would be obtained in another relevant cell type that does form these structures.

Can full scans of the Western blots be provided to see molecular size differences of the expressed isoforms?

Can the authors discuss whether the GFP tag is of potential influence for the structure of the Dp isoforms?

Reviewer 2

Advance summary and potential significance to field

In this study, the authors investigated how desmoplakin (DP) isoforms influence the spatial positioning of the DP tail domain within the desmosomal plaque. Using high-resolution microscopy and quantitative analysis, they demonstrated that the location of the DP tail domain is isoform-dependent and correlates with the length of the rod domain, while the DP head domain remains in a constant position across isoforms. Based on these findings, they suggested a variable tail location model, in which the DP isoforms were arranged with their rod domains parallel, at similar angles from the plasma membrane. These results provide insights into the role of DP isoforms in desmosomal architecture and function.

I recommend this manuscript be revised. I like how this study provides a novel perspective on the structural dynamics of desmosomes by linking DP isoform variability to functional architecture. However, several limitations need to be addressed.

Comments for the author

Major comments:

1. The quantitative analysis of the disperse cell adhesion assay (Fig. 1F) shows a large standard deviation (>100) in most samples, which suggests high variability in the data. This raises concerns about the reproducibility and statistical robustness of the measurements. Based on the methods, it seems that the authors use a pipette tip to apply mechanical stress to the cell sheets. Due to variations in manual pipette aspiration force, it is often difficult to generate uniform mechanical stress across samples. This might cause the high variability in the data. To ensure a more robust measurement, I would recommend that the authors use better-controlled alternative methods to apply mechanical stress in the disperse assay.

2. The authors need to critically choose statistical methods used in the data analysis. Their use of a one-way ANOVA as the statistical method for the analysis of all the data should be validated with a normality test as well as a variance test. For instance, given the small sample size in the disperse assay (<10), the assumption of normal data distribution within each group and equal variance becomes critical, as the test can be less reliable if the data does not meet these two criteria. Especially, given the outliers in their data, it would be better for the authors to consider a more appropriate statistical test for the analysis. After obtaining better-validated data, more explicit statements can then be made about the capacity of DP isoforms to resist mechanical strain (Page 5, line 8).

This also holds for the analysis of the scatter plot of head domain plaque-to-plaque distances since the p values suggest that the DP head domain position in the desmosomal plaque is NOT isoform independent (the p-values show a statistical significant difference between that the head domain

positions of DPI and the other isoforms). I suspect that this is because an inappropriate statistic is being used to calculate p-values.

3. The study assumes desmosomes have mirror symmetry across the midline (Page 5, line 30), which indicates that the "plaque-to-plaque" distance represents the average separation of mEGFP on the tail domain of DP across neighboring cells. However, is this assumption explicitly supported by prior literature? Since no citation is provided, this assumption needs further justification.

Alternatively, it would be a good idea to use a desmosomal cadherin extracellular domain antibody as a reference and measure the "plaque-to-midline" distance between the cadherin ectodomain and the tail domain as additional support for the analysis. This further validation would provide a more precise and unbiased midline marker for measuring plaque distances despite mirror symmetry.

4. The authors used rotary shadow electron microscopy to determine the angle of alignment of DP. However, experimental data was only collected for the DPI isoform but not for DPIa and DPII (Page 6, line 6). Instead, the authors indirectly estimate the DPIa and DPII length based on the correlation between the rod domain length and the amino acid sequence. While rod domain length does scale with amino acid count, this approach assumes a linear relationship, which may not fully account for structural flexibility or conformational changes. The fact that DPII exhibits a variety of conformations in measurements (Page 6, line 6) suggests that different isoforms might have different levels of flexibility, making an indirect estimation less accurate.

To better validate their approach, the authors need to either cite previous studies showing a strong correlation between rod domain length and amino acid number across DP isoforms or perform alternative experiments, such as cryo-EM or SAXS, to validate the estimated length.

Finally, confidence in the geometric model that the authors propose (Figure 3) will be improved if the authors can estimate error in the calculated DP angles. One way to do this would be to propagate errors from their experimental measurements into their angle calculation.

5. The study only examined one DP isoform at a time in cells, which helps explain the structure of each isoform but does not fully capture physiological conditions where multiple DP isoforms co-exist. One key question unaddressed question is if the desmosomal architecture is altered when two isoforms are co-expressed? Additional experiments that express two isoforms together and analyze whether the tail domain positioning remains isoform-dependent or if a hybrid structure form would improve the current model with more physiological relevance.

Minor comments:

1. It would be better to have another marker, such as a desmosomal cadherin, to indicate the location of cell-cell borders in Fig. 1D.

2. It is a little confusing how many data points are in each group shown in Fig 1F. Even though it is stated as 3 biological replicates with 3 technical replicates in the methods. It is better to specify the n in the figure legend as well.

Reviewer 3

Advance summary and potential significance to field

In this manuscript by Ainslie et al., the authors aimed to identify the architectural arrangement of distinct Desmoplakin (DP) isoforms. Utilizing dSTORM and DP KO HaCaT cells stably expressing DP isoforms, the authors demonstrate that the DP tail domains are different between isoforms and that this correlates with rod length. The authors propose a variable tail location model showing that the DP isoforms are arranged with an acute angle relative to the plasma membrane. While a paper from the same lab (Stahley et al., 2016) had already characterized desmosome structure by dSTORM, this manuscript is a nice supplement to this work by showing the difference in length and positioning of the isoforms. Nevertheless, this reviewer feels that a discussion of the biological significance of these findings and how this study moves the field forward is lacking.

Comments for the author

Major comments:

- 1) The authors should demonstrate that the C-terminal EGFP tag does not impact the DP isoform binding to intermediate filaments or confound the demonstrated resistance to mechanical strain (Fig. 1F).
- 2) Would be useful to include the raw images for each western blot (Fig. 1C) since the Westerns presented in the figure are very closely cropped - the kDa number labels appear arbitrary (they do not correspond to any known ladder).
- 3) In Fig. 2B, D, it is unclear why the 647 is located at the C versus N termini - authors need to improve their experimental design descriptions so the data can be better interpreted. It is also unclear what the N represents...plaque number? cell number? How many experimental and technical replicates are included in these graphs? A SuperPlot would help depict the variability in each N (Lord et al. JCB, 2020). Is the plaque-to-plaque distance consistent within the same desmosome?

Minor Comments:

- 1) Figure 2 would benefit from diagrams showing locations of each cell boundary and how the distances graphed in Fig. 2C were measured. The zoomed-out images show significant variability of the tail domain plaques, so how were the zoomed-in regions selected?
- 2) In Fig. 2A, how do the authors explain why the DP11 plaques are smaller/shorter?
- 3) For the constant tail location model, if the distance between the tails vary within the same junction or desmosome, wouldn't this model not stand?
- 4) The authors mention "Using the measured head and tail domain lengths (16 nm each)"...no citation or explanation is included to validate this measurement.
- 5) How do the authors explain the distinction between their claims of ~20 degrees orientation and the larger angles shown in the EM in Figure 1 of this paper?
https://www.sciencedirect.com/science/article/pii/S0955067407001251?ref=pdf_download&fr=RR-2&rr=9182d81f7970dd19
- 6) Authors mention that "The head and tail domain plaque-to-plaque measurements for each isoform were acquired from desmosomes at the same stage of maturation". What would the authors pose would happen under different maturation conditions?

First revisionAuthor response to reviewers' comments

Reviewer 1: The report provides evidence for different localization of Desmoplakin isoforms based on dSTORM superresolution microscopy. This may explain why the different isoforms interact differently at the plasma membrane interface of desmosomes.

The study is very interesting conceptually, and the imaging is of very high resolution and well-controlled. There are, however, some points that currently limit the potential impact of the finding.

Response: Thank you for your critique. We have conducted additional experiments as suggested which we feel strengthens the manuscript substantially. Each point is addressed below in detail.

Reviewer1 Comment1: In various cell types the desmoplakin isoforms might be co-expressed in cells. Can the authors test whether the intrinsic difference of the isoforms in plaque to plaque differences are maintained if DP1, DP1a and/or DP11 are co-expressed? This would answer the question whether the organization of one Dp isoform affects another and will test the provided model in figure 3.

Reviewer1 Response1: Thank you for this question. We have included **new experiments** and a **new Figure 3** (reproduced below for your convenience) to address this comment. We feel this substantially strengthens our manuscript and are excited by the results.

First, we transfected DP KO HaCaTs with both DPI-mEGFP and DPII-mCherry. Using NSPARC, a spatial array confocal which can achieve approximately 150nm x-y resolution, we determined the DPI tail extended further into the cell than the DPII tail. This is clear in the images which show a “wider” DPI-mEGFP signal where the individual plaques are resolvable with a “narrower” DPII-mCherry in between. This visual appearance is confirmed by linescan of intensity across a desmosome. To show this arrangement was consistent across many junctions, linescans from 63 desmosomes were averaged together clearly retaining DPI tail positioned deeper into the cytosol than the DPII tail. This imaging of DPI and DPII in individual junctions is exciting and supports the variable tail location of DP isoforms. This data is in the new **Figure 3 A- C**.

Second, we conducted dSTORM analysis of either DPI-mEGFP or DPII-mEGFP transfected into WT HaCaTs which express both DPI and DPII (see western blot in **Figure 1B**). We focused on DPI and DPII as they are the most abundant isoforms. Our analysis found a significant difference in the plaque-to-plaque distance between DPI-mEGFP and DPII-mEGFP in WT HaCaTs, with DPI extending further from the membrane than DPII. This data is in the new **Figure 3 D,E**.

We would like to note that the mean plaque-to-plaque distance for DPI and DPII tail domains was different between the WT and DP KO HaCaT background. We can not rule out that this could be a biological difference, where the isoforms can influence one another. Alternately, we think this is likely due to minor differences in the confluency of the cells or maturity of the desmosomes. Cell confluency and desmosome maturity have previously been shown to impact DP tail domain location (Stahley et al., *Journal of Cell Science*, 2016, Beggs et al., *Tissue Barriers*, 2022). We include reflection on these possibilities in the **Discussion** section.

New Figure 3. DP tail architecture is isoform dependent in cells expressing both DPI and DPII
A) Representative NSPARC images of DPI-mEGFP (cyan) and DPII-mCherry (magenta) in HaCaT DP KO cells (scale bar = 1 μ m). **B)** Intensity line scan through the boxed desmosome in Fig. 3A. **C)** Average relative intensity line scan of 63 desmosomes expressing DPI-mEGFP and DPII-mCherry. Error bars indicate 95% confidence interval. **D)** Representative dSTORM images of WT HaCaTs expressing DPI-mEGFP or DPII-mEGFP labeled with Alexafluor647 anti-GFP nanobody to image the tail domain (scale bar = 1 μ m). **E)** Scatter plot of DP tail domain plaque-to-plaque distance of DPI-mEGFP or DPII-mEGFP in WT HaCaTs, DPI-mEGFP: n = 51, 211 \pm 34 nm, DPII-mEGFP: n = 53, 152 \pm 23 nm (mean \pm s.d). n is the number of desmosomes which were imaged from at least 2 independent experiments. Shapiro-Wilks test for normality confirmed data was normally distributed (p > 0.05 for each isoform). Unpaired t-test revealed there is a significant difference in the tail domain plaque-to-plaque distance between DPI and DPII-mEGFP in WT HaCaTs (**** p < 0.0001).

R1C2: *The expression of the Dp isoforms induces desmosome formation in these KO HaCaTs, however, one wonders what results would be obtained in another relevant cell type that does form these structures.*

R1R2: Thank you for this question. In this work, DP KO HaCaTs were employed because they allowed us to examine each isoform independent of one another. The new experiments we conducted in response to **R1C1** (described above) include quantification of DPI-mEGFP and DPII-mEGFP expressed in WT HaCaT cells. We found the variable tail location model was conserved in the presence of multiple DP isoforms, strengthening our conclusion.

While repeating these experiments in multiple cell lines was not feasible given our current resources, in the past we have found the nanoscale architecture of “core” desmosomal proteins, including desmoplakin, to be well conserved between cell types. Our lab has published DP architecture in primary human keratinocytes, A431 cells (human epidermal), MDCK cells (dog kidney, simple epithelia), HUC (human urothelial cells; transitional epithelia), and normal human epidermis. We see similar architecture in all (Stahley et al., *Journal of Cell Science*, 2016, Beggs et al., *Tissue Barriers*, 2022). Interestingly, we reported a change in DP tail localization based on desmosome maturity in primary HKs, A431, MDCK, and HUCs and changes in the location of the DP tail domain in hyper-adhesion (primary HKs) and in basal vs supra basal cells (human epidermis). Finally, unpublished data in the lab shows a change in DPI tail domain localization in hyperadhesive vs calcium dependent HaCaTs. Together these data demonstrate the ability of DP to take on different architectures. As is evident in the literature, desmosome ultrastructure is consistent between cell lines, again strengthening our choice of model. We have included a discussion of this in the manuscript.

R1C3: *Can full scans of the Western blots be provided to see molecular size differences of the expressed isoforms?*

R1R3: We have included larger crops of the western blots in **Figure 1B** and provided the full western blots in **new Figure S2**.

R1C4: *Can the authors discuss whether the GFP tag is of potential influence for the structure of the Dp isoforms?*

R1R4: Thank you for this important question. GFP was inserted at the C-terminus of DP with a 6 AA linker (DPPVAT), identical to that used in previous studies. We show that the GFP tagged isoforms localize at desmosomes, colocalize with Dsg3, restore ultrastructure, and increase resistance to mechanical strain compared to the DP KO HaCaT cell line. DP with a C-term fluorescent protein tag has been previously used in numerous functional studies. GFP tagged DP was able to facilitate desmosome assembly and has been used in functional assembly assays (Godsel et al., *Journal of Cell Biology*, 2005). DP-mEGFP was used by co- Author N. Bharathan to study desmosome fusion and fission dynamics (Bharathan et al., *Nature Cell Biology*, 2023). Co-author V. Spindler showed DP-mEGFP expressed in DP KO HaCaTs localizes to borders and promotes cadherin clustering (Wanuske et al., *Acta Physiologica*, 2020). DP S2849 promotes hyper-adhesion by enhancing the DP-keratin binding interaction. DP S2849G tagged with a C-terminal GFP promotes hyper- adhesion (Hobbs et al., *Journal of Investigative Dermatology*, 2012, Bartle et al., *Journal of Cell Biology*, 2020, Godsel et al., *Journal of Cell Biology*, 2005) and block Pemphigus Vulgaris autoantibody-mediated loss of cell cohesion (Dehner et al, *American Journal of Pathology*, 2014). Keratin filaments have been shown to nucleate and grow from DPI-mApple (mApple is a red fluorescent protein) (Moch et al, *Cellular and Molecular Life Science*, 2019). This multitude of data indicates that a C-terminal GFP tag does not impede DP function. We have added a of the manuscript on this topic.

Reviewer 2: *In this study, the authors investigated how desmoplakin (DP) isoforms influence the spatial positioning of the DP tail domain within the desmosomal plaque. Using high-resolution microscopy and quantitative analysis, they demonstrated that the location of the DP tail domain is isoform-dependent and correlates with the length of the rod domain, while the DP head domain remains in a constant position across isoforms. Based on these findings, they suggested a variable tail location model, in which the DP isoforms were arranged with their rod*

domains parallel, at similar angles from the plasma membrane. These results provide insights into the role of DP isoforms in desmosomal architecture and function.

I recommend this manuscript be revised. I like how this study provides a novel perspective on the structural dynamics of desmosomes by linking DP isoform variability to functional architecture. However, several limitations need to be addressed.

Response: We thank the reviewer for their feedback which we feel has significantly strengthened the manuscript. Our detailed responses are below.

R2C1: *The quantitative analysis of the dispase cell adhesion assay (Fig. 1F) shows a large standard deviation (>100) in most samples, which suggests high variability in the data. This raises concerns about the reproducibility and statistical robustness of the measurements. Based on the methods, it seems that the authors use a pipette tip to apply mechanical stress to the cell sheets. Due to variations in manual pipette aspiration force, it is often difficult to generate uniform mechanical stress across samples. This might cause the high variability in the data. To ensure a more robust measurement, I would recommend that the authors use better-controlled alternative methods to apply mechanical stress in the dispase assay.*

R2R1: Thank you for this comment. We agree with the reviewer on the methodology in the dispase assay and have repeated these experiments using a different mode of inducing fragmentation. In new experiments we used an automated pipette (Eppendorf Xplorer) as has been previously published by co-author V. Spindler (see for example Dehner et al., *The American Journal of Pathology*, 2014, Schlögl et al., *Frontiers in Immunology*, 2018). Because the fluid handling is automated, the stress is applied with equal force for each pipette. We would like to note that we attempted to identify other methods of fragmentation and tested the many orbital shakers and tube rotators available in our department. None were able to fragment the WT HaCaTs and therefore were not appropriate for our experiment.

Shapiro-wilks test for normality confirmed all samples were normally distributed ($p > 0.05$). Brown-Forsythe test determined the variance was not equal between groups ($p < 0.05$). Brown-Forsythe and Welch ANOVA with Dunnett's t3 multiple comparison test was used to assess the DP tail domain position. This data is in the **new Figure 1 E, F** and reproduced here for your convenience.

We note in this data there is larger variability in the DP KO sample. A similar fragment distribution in the DP KO HaCaTs was shown in co-author V. Spindler's manuscript (see Fig 1b in Wanuske et al., *Acta Physiologica*, 2020). We feel the overall fragility of this cell monolayer and the resulting large number of fragments, some of which may be below our detection limit, is the underlying reason

New Data Figure 1 E) Number of fragments from a dispase cell adhesion assay for WT, DP KO, and DP-mEGFP isoform HaCaT cells (mean \pm s.d., $n = 9$ with 3 cell sheets in each of 3 biological replicates, each replicate with a unique symbol shape) WT HaCaT: 16 ± 8 , DP KO HaCaT: 424 ± 154 , DPI-mEGFP: 200 ± 56 , DPIa-mEGFP: 149 ± 56 , DP II-mEGFP: 201 ± 51 . Shapiro-Wilk test confirmed normal distribution ($p > 0.05$). Brown-Forsythe test determined the variance was not equal between groups ($p < 0.05$). Brown-Forsythe and Welch ANOVA with Dunnett's t3 multiple comparison test assessed the statistically significant differences (**** $p < 0.0001$, *** $p < 0.001$, ** $p < 0.01$, * $p < 0.05$). **F)** Representative images from the dispase cell adhesion assay. The key conclusion drawn from these data is that expression of the GFP tagged isoforms provides cells with resistance to mechanical strain over the DP KO.

R2C2: The authors need to critically choose statistical methods used in the data analysis. Their use of a one-way ANOVA as the statistical method for the analysis of all the data should be validated with a normality test as well as a variance test. For instance, given the small sample size in the dispase assay (< 10), the assumption of normal data distribution within each group and equal variance becomes critical, as the test can be less reliable if the data does not meet these two criteria. Especially, given the outliers in their data, it would be better for the authors to consider a more appropriate statistical test for the analysis. After obtaining better-validated data, more explicit statements can then be made about the capacity of DP isoforms to resist mechanical strain (Page 5, line 8). This also holds for the analysis of the scatter plot of head domain plaque-to-plaque distances since the p values suggest that the DP head domain position in the desmosomal plaque is NOT isoform independent (the p -values show a statistical significant difference between that the head domain positions of DPI and the other isoforms). I suspect that this is because an inappropriate statistic is being used to calculate p -values.

R2R2: Thank you for your assistance and suggestions - we apologize for failing to properly report all statistical tests. All figure legends have been updated with appropriate details. For dSTORM data (Fig. 2C, E and new Fig. 3C) we tested for normality using the Shapiro-Wilk test and equal variance using the Brown-Forsythe test.

This is now noted in the figure legends. The DP head domain data (Fig. 2E) passed the tests for normality and equal variance and was analyzed by a One-way ANOVA with Tukey's multiple comparison post-hoc test. The DP tail domain data in Fig. 2C passed the normality test but failed the equal variance test. We conducted a Brown-Forsythe and Welch ANOVA with a Dunnett's t3 multiple comparison post-hoc test. For the tail domain data in WT HaCaTs a Student's t-test (Fig. 3C) was performed as the data passed both the normality and equal variance tests.

The reviewer is correct the DP head domain plaque-to-plaque distance data is statistically significant. Indeed, we used the isoform specific measurements when calculating the proposed angles of alignment. While we do not think that these differences are likely to be biologically relevant, this phrasing was not appropriate and has been removed. The salient feature is that the largest variation in mean head domain position between isoforms is 10nm, which cannot account for the over 60nm difference in the mean tail domain plaque-to-plaque distances. Therefore differences in tail domain plaque-to-plaque distance can not be explained by different overall position of the DP isoforms. We have clarified this in the text, which now reads as follows:

“The relatively minor 10 nm differences while statistically significant, are not sufficient to explain the 68 nm difference in the tail domain plaque-to-plaque distance.”

The disperse fragmentation assay was repeated using a different method of inducing fragmentation as described above. A Shapiro-Wilk test determined the data was normally distributed ($p > 0.05$). The data was determined to not have equal variance by the Brown-Forsythe test ($p < 0.05$). We then conducted a Brown-Forsythe and Welch ANOVA with a Dunnett's t3 multiple comparison post-hoc test.

R2C3: *The study assumes desmosomes have mirror symmetry across the midline (Page 5, line 30), which indicates that the "plaque-to-plaque" distance represents the average separation of mEGFP on the tail domain of DP across neighboring cells. However, is this assumption explicitly supported by prior literature? Since no citation is provided, this assumption needs further justification. Alternatively, it would be a good idea to use a desmosomal cadherin extracellular domain antibody as a reference and measure the "plaque-to-midline" distance between the cadherin ectodomain and the tail domain as additional support for the analysis. This further validation would provide a more precise and unbiased midline marker for measuring plaque distances despite mirror symmetry.*

R2R3: The mirror symmetry of desmosomes has been a vital element to desmosomal studies and noted since early electron micrographs were available (Bharathan et al., 2023). Odland in 1958 presented a highly resolved electron density pattern of the desmosome at the cell-cell contact. Desmosomes had five dense stripes, with a thickness of ~ 500 Å. Densitometric analysis revealed the desmosome is symmetrical and comprises an IDP and ODP in each cell, appearing as mirror images on either side of the cell-cell contact. Later studies described similar desmosome ultrastructure in other tissues and species, including frog mesothelium and rat and guinea pig stomach, jejunum, and colon epithelia (Hama, 1960; Farquhar and Palade, 1963). Regions with similar electron density patterns are observed in the intercalated disc of the cardiac muscle, known to contain elements of both adherens junctions and desmosomes (Sjöstrand and Andersson, 1954; Muir, 1957; Franke et al., 2006).

We have previously utilized the mirror symmetry of desmosomes to facilitate their characterization (Stahley et al., *Journal of Cell Science*, 2016). While North et al. (*Journal of Cell Science*, 1999) reported measurements to the midline, the "side" of a desmosome each measurement came from was not noted and were averaged together. We have added citations in the manuscript to these points.

The use of cadherins as an extracellular marker in dSTORM to measure the "plaque-to-midline" is an interesting suggestion. Due to challenges with multi-color super-resolution, this experiment was outside the scope of this manuscript. However, this is something that we believe we could incorporate into future work address desmosome symmetry in super-resolution directly.

R2C4: *The authors used rotary shadow electron microscopy to determine the angle of alignment of DP. However, experimental data was only collected for the DPI isoform but not for DPIa and DPII (Page 6, line 6). Instead, the authors indirectly estimate the DPIa and DPII length based on the correlation between the rod domain length and the amino acid sequence. While rod domain length does scale with amino acid count, this approach assumes a linear relationship, which may not fully account for structural flexibility or conformational changes. The fact that DPII exhibits a variety of conformations in measurements (Page 6, line 6) suggests that different isoforms might have different levels of flexibility, making an indirect estimation less accurate. To better validate their approach, the authors need to either cite previous studies showing a strong correlation between rod domain length and amino acid number across DP isoforms or perform alternative experiments, such as cryo-EM or SAXS, to validate the estimated length.*

Finally, confidence in the geometric model that the authors propose (Figure 3) will be improved if the authors can estimate error in the calculated DP angles. One way to do this would be to propagate errors from their experimental measurements into their angle calculation.

R2R4: We apologize for the confusion. The DP rotary shadow EM we refer to is previously published (O'Keefe et al., *Journal of Biological Chemistry*, 1989,) and is the only, to our knowledge, data measuring the length of DP or the DP rod domain. We note that when we

attempted to fold DP AlphaFold, for a monomer, a single continuous alpha helix was generated for the rod. For a DP dimer, AlphaFold generated a knot in the rod domains, which is a very unusual fold considered highly unlikely. We agree with the reviewer our model may not account for structural flexibility or conformational changes and have included this caveat in the discussion section.

We agree additional structural information about DP would benefit not only this work, but the field as a whole. This is however beyond the scope of this manuscript and, unfortunately, beyond the current funding limits of the lab. We have significantly revised the discussion to include alternate hypotheses as to how DP is arranged to arrive at the variable tail location and additional support for our proposed model.

Following the reviewer's excellent suggestion, we propagated the error to through our calculation of DP angle and included this information in the figure and text: DPI: $22 \pm 6^\circ$, DPIa: $25 \pm 8^\circ$, and DP11; $21 \pm 9^\circ$.

R2C5: *The study only examined one DP isoform at a time in cells, which helps explain the structure of each isoform but does not fully capture physiological conditions where multiple DP isoforms co-exist. One key question unaddressed question is if the desmosomal architecture is altered when two isoforms are co-expressed? Additional experiments that express two isoforms together and analyze whether the tail domain positioning remains isoform-dependent or if a hybrid structure form would improve the current model with more physiological relevance.*

R2R5: Thank you for this important experimental suggestion. We conducted two new sets of experiments which show the DP architecture has an isoform dependent variable tail location when multiple DP isoforms are expressed. These experiments are the **new Figure 3**. This was show by: 1 Co-expressing DPI-mEGFP and DP11-mCherry in DP KO HaCaTs and imaging by NSPARC and 2 Expressing DPI-mEGFP or DP11-mEGFP separately in WT HaCaTs, which express both DPI and DP11, and quantifying DP tail architecture by dSTORM. Please see our response to **Reviewer 1 Comment 1** for more details and the reproduced figure.

Minor Comments (MC)

R2MC1: *It would be better to have another marker, such as a desmosomal cadherin, to indicate the location of cell-cell borders in Fig. 1D.*

R2MR1: Thank you for the suggestion. We have updated our images, now **Figure 1C**, to include staining for DSG3.

R2MC2: *It is a little confusing how many data points are in each group shown in Fig 1F. Even though it is stated as 3 biological replicates with 3 technical replicates in the methods. It is better to specify the n in the figure legend as well.*

R2MR2: We apologize for this oversight. We updated the figure legends to explicitly state n and the number of biological and technical replicates. In **Figure 1E** each biological replicate has been assigned a unique shape to streamline data interpretation.

Reviewer 3: *In this manuscript by Ainslie et al., the authors aimed to identify the architectural arrangement of distinct Desmoplakin (DP) isoforms. Utilizing dSTORM and DP KO HaCaT cells stably expressing DP isoforms, the authors demonstrate that the DP tail domains are different between isoforms and that this correlates with rod length. The authors propose a variable tail location model showing that the DP isoforms are arranged with an acute angle relative to the plasma membrane. While a paper from the same lab (Stahley et al., 2016) had already characterized desmosome structure by dSTORM, this manuscript is a nice supplement to this work by showing the difference in length and positioning of the isoforms. Nevertheless, this reviewer feels that a discussion of the biological significance of these findings and how this study moves the field forward is lacking.*

Response: Thank you for your comments on our manuscript. We have enhanced the discussion section of the manuscript to highlight the biological significance of these findings, and how this

work moves the field forward. Our responses to specific comments can be found below.

R3C1: The authors should demonstrate that the C-terminal EGFP tag does not impact the DP isoform binding to intermediate filaments or confound the demonstrated resistance to mechanical strain (Fig. 1F).

R3R1: Thank you for this comment. This is an important concern, please see our response above to Reviewer 1 Comment 4 (R1C4) for a detailed discussion.

R3C2: Would be useful to include the raw images for each western blot (Fig. 1C) since the Westerns presented in the figure are very closely cropped - the kDa number labels appear arbitrary (they do not correspond to any known ladder).

R3R2: Thank you for the comment, we apologize for the confusion. The kDa numbers indicated were the known kDa of the proteins blotted for, and not the ladder. We have remade the western blot figure (now **Figure 1B**) with larger crops and indications only of the ladder molecular weights. In addition, all blots are now included in **new Figure S2**.

R3C3: In Fig. 2B, D, it is unclear why the 647 is located at the C versus N termini - authors need to improve their experimental design descriptions so the data can be better interpreted. It is also unclear what the N represents...plaque number? cell number? How many experimental and technical replicates are included in these graphs? A SuperPlot would help depict the variability in each N (Lord et al. JCB, 2020). Is the plaque-to- plaque distance consistent within the same desmosome?

R3R3: Thank you for the suggestion. The location of Alexafluor 647 corresponds to the labeling in each experiment. In **Figure 2B**, we used an anti-GFP nanobody conjugated with Alexafluor 647. This binds EGFP and allow us to quantify the C-term architecture. In **Figure 2C**, we used an anti-DP rabbit primary antibody that was raised against the head domain of DP and an anti-rabbit Alexafluor 647 secondary antibody. This allows us to quantify the N-term architecture. This labeling strategy has been clarified in the text as well as the figure legend. The antibodies and their uses have been more fully described in the methods.

In the figure legend, n refers to number of desmosomes analyzed. There was a minimum of three independent experiments for each isoform. These values are now indicated in the figure legend. Each data point represents the average plaque-to-plaque distance for an individual desmosome. We agree it would be interesting to examine if there is variability within individual junctions, however this would address a different question than the one we set out to answer. In this work, we measure the plaque to plaque distance from an average linescan taken across the entire length of the desmosome (see figure below), which eliminates intra- desmosomal variability, as previously published (Stahley et al, *Journal of Cell Science*, 2016, Beggs et al, *Tissue Barriers*, 2020, Beggs et al., *Methods in Molecular Biology*, 2021). This is illustrated in a **new Figure S1**, reproduced below.

Supplemental Figure 1: dSTORM Analysis Pipeline: A) Overlay of widefield and dSTORM

reconstructed image of DPI-mEGFP HaCaT Cells. **B)** Cropped reconstructed desmosome with yellow arrows indicating the linescan used to measure the plaque-to-plaque (PtP) distance. **C)** dSTORM analysis output graph of intensity as a function of distance along the linescan. The distance between the two peaks is the PtP distance. **D)** Schematic representation of PtP distance.

Minor Comments (MC)

R3MC1: Figure 2 would benefit from diagrams showing locations of each cell boundary and how the distances graphed in Fig. 2C were measured. The zoomed-out images show significant variability of the tail domain plaques, so how were the zoomed-in regions selected?

R3MR1: We apologize for the confusion. The new **Figure S1** (reproduced above) illustrates the analysis pipeline for the dSTORM images. The images in 2B and 2D are representative individual desmosomes, while each data point on the plots in 2C and 2E is one individual desmosome. The architecture is not the same across all desmosomes and instead there is a range of plaque-to-plaque distances measured from desmosomes expressing each isoform. We have provided 3 example desmosomes from each DP isoform from our dSTORM analysis pipeline below. Here you see the desmosome (bottom) and linescan (top) the red arrow heads indicate the center of the plaques used to calculate the plaque-to-plaque distance. Here you can see more of the diversity within and across isoforms.

R3MC2: In Fig. 2A, how do the authors explain why the DPII plaques are smaller/shorter?

R3MR2: This is an interesting observation. While we did not quantify plaque length in this work previously we found there is heterogeneity in plaque length measured by dSTORM (Stahley et al 2016), much like there is heterogeneity in desmosome length, which has been shown to range between several hundred nm to microns by EM a (see for example Hino et al., *Acta Dermatovener*, 1989, Odland 1958, Tucker et al, 2014). Based on our proposed model, it is possible the plaque length for DPII are on average shorter than DPI based on the architecture - DPI being longer, can splay outwards further generating longer plaques. To make this statement with confidence, both DPI and DPII should be measured in the same junction from multi-color super-resolution images. Unfortunately, this was not feasible given our current set up and is why we choose not to draw any conclusions about desmosome length.

R3MC3: 3) For the constant tail location model, if the distance between the tails vary within the same junction or desmosome, wouldn't this model not stand?

R3MR3: Thank you for the thoughtful comment. The two models we put forward admittedly represent 2 extremes to illustrate our hypothesis - many permeations are possible in a spectrum between the two models. For example as the reviewer suggested, maybe there is heterogeneity within a single junction with variable DP tail domain locations. Another example would be a variable tail location, but with DP isoforms at different angles. Upon reflection we have updated the terminology in the manuscript regarding our model. We describe our data and showing a “Variable tail location” and propose the “aligned angle” mode to describe how DP could be arranged. We have expanded the discussion to include alternate models of DP architecture and place our results in context of the field. There is also an expanded discussion to include alternate models and interpretations.

R3MC4: *The authors mention "Using the measured head and tail domain lengths (16 nm each)"...no citation or explanation is included to validate this measurement.*

R3MR4: Thank you for pointing out this error in our submission. We have now included the appropriate citation (O’Keefe et al., 1989).

R3MC5: *How do the authors explain the distinction between their claims of ~20 degrees orientation and the larger angles shown in the EM in Figure 1 of this paper?*
https://www.sciencedirect.com/science/article/pii/S0955067407001251?ref=pdf_download&fr=RR-2&rr=9182d81f7970dd19

R3MR5: The paper the reviewer refers to, *Desmosomes from a Structural Perspective* by D. Stokes (2007) is a very nice and thought provoking review. In Figure 1, proteins are represented by a combination of structures and spheres and overlaid onto an EM micrograph to illustrate a possible architecture. Desmoplakin was represented as described by the authors as folded over on itself in 3 separate folds with presumably flexible linkers in between. Importantly, the structure used here is not of DP rod. Quoted from the figure legend: “*The central domain of desmoplakin is predicted to form a long coiled coil consisting of 889 residues and is shown here as three discontinuous lengths derived from the X-ray structure of tropomyosin.*” The author later describe the impossibility of a fully extended desmoplakin, in agreement with our work. The hypothesis put forward in this review is that the central domain of DP is folded within the IDP. The author states “*The N-terminal and C-terminal domain can be expected to be globular with a dimension of 5-10 nm each, suggesting that the central, α -helical domain must be folded up within the IDP. One might speculate that such folding would provide an extensibility to the IDP, which could be useful in maintaining cell adhesion when the tissue is under shear stress.*” This is also one of several hypotheses put forward by North et al 1999 when they reached the same conclusion about the impossibility of a fully extended DP fitting with their immunogold EM work.

So why do we propose an extended rod sitting at an angle?

First, there is no structural evidence that the DP rod can be folded onto itself. OKeefe doesnt see this. But there is a huge gap in our structural knowledge here. *The field still does not have a molecular structure of full length desmoplakin.* When we attempted to fold DP alphafold, for a monomer, a single continuous alpha helix was generated for the rod. For a DP dimer, alphafold generated a knot in the rod domains, which is a very unusual fold we considered highly unlikely. *Additional structural studies of the desmosome component proteins will be needed to fully bridge the structure/function gap.*

Second, in the initial manuscript from our lab using super-resolution to investigate desmosome architecture we found the average desmosome length, parallel to the plasma membrane, was significantly larger for the DP C- term than the DP N-term. The length of the N-term closely matched that of the other ODP protein components including Dsg3-cterm and plakoglobin. While confusing at first, we concluded that this suggested DP was splaying outward as a result of sitting at an acute angle in the plaque, with the C-term generating a larger keratin interaction interface.

Third, we have previously shown DP C-term can adopt a range of positions based on maturity of the junction, suggesting a more analog continuous tuning. We hypothesize that this ability to adopt multiple positions in the IDP would be provided by a change in angle. Alternately straightening/unfolding of the rod domain would produce discrete digital positions. Of course, a

combination of folded and unfolded within a single junction cannot be ruled out.

We have included a new paragraph of the discussion section, discussing these different hypotheses.

R3MC6: *Authors mention that "The head and tail domain plaque-to-plaque measurements for each isoform were acquired from desmosomes at the same stage of maturation". What would the authors pose would happen under different maturation conditions?*

R3MR6: Thank you for the insightful question. It has been established that not only do subunit isoforms of desmosomes change, the architecture of the tail domain of DP to change during desmosome maturation. We have shown that as desmosomes mature and eventually become hyper-adhesive, the tail domain of DP increases its proximity to the plasma membrane (Beggs et al, Tissue Barriers, 2022, Stahley et al., *Journal of Cell Science*, 2016). We included these ideas in the discussion section.

Additionally, we have unpublished data that shows each the proximity of the tail domain for each isoform to the plasma membrane correlates to desmosome maturation. The proximity to the plasma membrane is a gradual change and takes days for the gradual change.

Second decision letter

MS ID#: jcs.263906R1

MS TITLE: Desmoplakin tail domain position in the desmosomal plaque is isoform dependent

AUTHORS: Collin M Ainslie; Krishna Patel; Yen T. B. Tran; Navaneetha Krishnan Bharathan; Samuel C. Bartley; Volker Spindler; Alexa L Mattheyses

ARTICLE TYPE: Short Report

Dear Alexa,

We have now reached a decision on the above manuscript.

To see the reviewers' reports and a copy of this decision letter, please go to:

As you will see, the reviewers gave favourable reports but raised some critical points that will require amendments to your manuscript. I hope that you will be able to carry these out because I would like to be able to accept your paper, depending on further comments from reviewers.

Reviewer 1

Advance summary and potential significance to field

The report provides very convincing evidence for different localization of Desmoplakin isoforms based on dSTORM superresolution microscopy. This explains why the different isoforms interact differently at the plasma membrane interface of desmosomes.

The study is very interesting conceptually, and the imaging is of very high resolution and well-controlled.

Comments for the author

The authors have done a great job in addressing the reviewers questions, and the manuscript is much stronger now by including WT Hacats and co expression and imaging of the desmoplakins.

Reviewer 2*Advance summary and potential significance to field*

The authors have done an admirable job with their revisions. They have now incorporated a better-controlled methodology for the dispase assay and added normality and variance tests for their statistical analysis. The results reflect their key conclusion, that expression of the GFP-tagged isoform provides cells with resistance to mechanical strain over the DP KO.

I have one suggestion that I believe the authors should incorporate before this manuscript is published. On the first page of the results section (line 50), the authors state that "Interestingly, while there was no significant difference in fragmentation between isoforms, indicating a similar capacity to resist mechanical strain,.....". This statement is not supported by the results, since no statistical tests are shown between DP isoforms in the new Figure 1E. Based on the mean value provided in the figure legend, the DP Ia with 149 mean fragments seems to significantly differ from the DP I and DP II with ~ 200 mean fragments. The authors therefore need to demonstrate the statistical significance between the DP isoforms in the dispase assay and modify the statement accordingly based on the results.

Reviewer 3*Advance summary and potential significance to field*

Minor comments

The authors did a thorough job addressing the requested revisions and clarifications. The manuscript is deemed acceptable for publication should the minor changes below be addressed:

Westerns:

- * Authors should clearly label the sizes of each protein in each Western blot image (preferably on the opposite side of the ladder for clarity) and/or in the corresponding figure legends (new Figs 1B and S2).
- * The n for each western is not specified.
- * Would be useful to include quantification of expression levels, particularly since the authors mention in the manuscript "This difference likely arises from a difference in DP expression level" (new lines 53-54).

Abstract:

- * This sentence of the abstract makes it unclear whether this model has already been proposed and the data in the manuscript supports it, or whether this manuscript establishes an entirely novel model. Please rephrase to clarify.

"We propose the aligned angle model, with each DP isoform co-aligned at an acute angle relative to the plasma membrane" (new lines 18-19).

- * The authors could more directly state the "insight" their results reveal for clarity and impact:

"These results provide valuable insight into DP isoform architecture and desmosome function" (new lines 19-20).

Methods:

- * Confluency levels should be specified, as the authors mention:

"We note that the mean plaque-to-plaque distance for DP I and DP II tail was different between the WT and DP KO HaCaT background. ...Alternately, it could result from minor differences in the cell

confluency which impacts desmosome maturity, both of which have previously been shown to impact DP tail domain location" (new lines 10-16 pg.)

* Image Analysis and Immunofluorescence sections should contain a concise but detailed description and not require that readers look up a previous paper.

* Statistical Analysis section should specify the individual tests that were applied.

Second revision

Author response to reviewers' comments

Reviewer 1: SUMMARY OF THE ADVANCE MADE IN THIS PAPER AND ITS POTENTIAL SIGNIFICANCE TO THE FIELD The report provides very convincing evidence for different localization of Desmoplakin isoforms based on dSTORM superresolution microscopy. This explains why the different isoforms interact differently at the plasma membrane interface of desmosomes.

The study is very interesting conceptually, and the imaging is of very high resolution and well-controlled.

R1C1: The authors have done a great job in addressing the reviewers questions, and the manuscript is much stronger now by including WT Hacats and co expression and imaging of the desmoplakins.

Response 1: Thank you for your feedback and initial comments that were essential in strengthening this study.

Reviewer 2: The authors have done an admirable job with their revisions. They have now incorporated a better-controlled methodology for the dispase assay and added normality and variance tests for their statistical analysis. The results reflect their key conclusion, that expression of the GFP-tagged isoform provides cells with resistance to mechanical strain over the DP KO.

Response: Thank you for the compliment on the revision. We have addressed your additional comments below.

R2C1: I have one suggestion that I believe the authors should incorporate before this manuscript is published. On the first page of the results section (line 50), the authors state that "Interestingly, while there was no significant difference in fragmentation between isoforms, indicating a similar capacity to resist mechanical strain,.....". This statement is not supported by the results, since no statistical tests are shown between DP isoforms in the new Figure 1E. Based on the mean value provided in the figure legend, the DPl_a with 149 mean fragments seems to significantly differ from the DPl and DPl_{II} with ~ 200 mean fragments. The authors therefore need to demonstrate the statistical significance between the DP isoforms in the dispase assay and modify the statement accordingly based on the results.

R2R1: Thank you for bringing this to our attention. When conducting the statistical analysis we did include comparisons between all groups (WT, DP KO, DPl, DPl_a, and DPl_{II}), however "ns" was not indicated on the plot. We now updated the **manuscript, Figure 1E, Figure 1 legend** to reflect this.

Reviewer 3 Minor comments The authors did a thorough job addressing the requested revisions and clarifications. The manuscript is deemed acceptable for publication should the minor changes below be addressed:

Response: Thank you for the feedback, please see our detailed responses to the additional questions below.

Westerns

R3C1: * Authors should clearly label the sizes of each protein in each Western blot image (preferably on the opposite side of the ladder for clarity) and/or in the corresponding figure legends (new Figs 1B and S2).

R3R1: Thank you for the suggestion regarding labeling of our western blots. To address, we have updated the legends of **Figure 1 and Figure S2** to include the molecular weight of WT DP, each DP-mEGFP isoform, and GAPDH. The predicted molecular weights are: DPI, 332 kDa; DPII 260 kDa ;DPI-mEGFP, 359 kDa; DPIa-mEGFP 306 kDa; DPII, 287 kDa; and GAPDH,37 kDa.

R3C2: * The n for each western is not specified.

R3R2: Thank you for bringing this to our attention. We have updated the text and **Figure S2 legend**. 4 western blots contain DP-mEGFP isoform expressing HaCaTs and 3 WT and DP KO HaCaTs.

New Figure S2 legend

“A) Immunoblot of DP (left) and GAPDH (right) from DP-mEGFP HaCaT and DP KO HaCaT lysates. **B)** Immunoblot probing for DP (left) and GAPDH (right) from WT and DP KO HaCaT lysates **C)** Immunoblot for DP from DP-mEGFP HaCaT DP KO and WT HaCaT lysates. **D)** Immunoblot for DP from DP-mEGFP HaCaT DP KO lysates. **E)** Immunoblot of DP from DP- mEGFP HaCaT DP KO and WT HaCaT lysates. A total of 5 western blots are presented. 4 blots show DP-mEGFP isoforms (A, C, D, E) and 3 blots show WT and KO HaCaTs (B, C, E). Predicted molecular weights: DPI-mEGFP 359 kDa, DPIa-mEGFP 306 kDa, DPII-mEGFP 287 kDa, DPI 332 kDa, DPII 260 kDa, and GAPDH 37 kDa. All blots use antibodies indicated within Materials and Methods.”

R3C3: * Would be useful to include quantification of expression levels, particularly since the authors mention in the manuscript "This difference likely arises from a difference in DP expression level" (new lines 53-54).

R3R3: Thank you for this suggestion. We had based this comment on our observations from the Western Blots. However, upon reflection this point is not central to our conclusions. Due to challenges in quantification, we have elected to remove this line from the manuscript. We feel this is appropriate because the variable tail locations of the DP isoforms were not dependent on protein level as similar architecture was present in both DP KO and WT cells. Additionally, all architectural measurements were made on individual desmosomes, the protein content of which may or may not reflect a population average. The key result from the disperse fragmentation assay is that re-introduction of DP-mEGFP isoforms to DP KO HaCaTs provides protection against mechanical force when compared to the DP KO HaCaTs. The connection between isoforms and expression level may be more complex as found in Cabral et al., (2012). In this study, siRNA KD of DPI or DPII led to differences in adhesion. We added some discussion of this relationship to the discussion section: *“Additionally, the disperse fragmentation assay raises interesting questions regarding DP isoform dependence in the resistance to mechanical strain. While the data presented here suggests there is no significant difference between isoforms, differences in adhesive function between DPI and DPII have been demonstrated by Cabral et al., (2012) using a unique siRNA approach. Additional work is needed to fully understand how and why desmosomes provide mechanical strength to cells.”*

Abstract

R3C4: * This sentence of the abstract makes it unclear whether this model has already been proposed and the data in the manuscript supports it, or whether this manuscript establishes an entirely novel model. Please rephrase to clarify.

"We propose the aligned angle model, with each DP isoform co-aligned at an acute angle relative to the plasma membrane" (new lines 18-19).

R3R4: Thank you for your suggestion. We have rephrased the sentence to make clear we are proposing a new model of desmoplakin architecture. The sentence now reads:

"We propose a novel aligned angle model, with each DP isoform co-aligned at an acute angle relative to the plasma membrane."

R3C5: * The authors could more directly state the "insight" their results reveal for clarity and impact:

"These results provide valuable insight into DP isoform architecture and desmosome function" (new lines 19-20).

R3R5: Thank you for this feedback. We note we are limited on word count (180) in the abstract but have revised to text to indicate more clearly the insights gained from this work *"These results provide insight into how DP architecture supports desmosome function."*

Methods:

R3C6: * Confluency levels should be specified, as the authors mention:

"We note that the mean plaque-to-plaque distance for DPI and DPII tail was different between the WT and DP KO HaCaT background. ...Alternately, it could result from minor differences in the cell confluency which impacts desmosome maturity, both of which have previously been shown to impact DP tail domain location" (new lines 10-16 pg.)

R3R6: Thank you for your suggestion. Samples were fixed between 1-2 days confluent and we have updated our methods to include this in the "immunofluorescence" section. While in general the KO cells were closer to 1 day and the WT cells closer to 2 days confluent, we do not have the time resolution to make a conclusive statement. Us and others have noted that desmosomes in cells at the edge of a colony have a less mature morphology than those in the center of a cell colony. Additionally it is well documented that desmosomes "switch" from calcium dependent to calcium independent after 5 days in confluent culture. Together, this indicates confluency plays a role in desmosome structure. In the discussion we feel it is important to bring this possibility forward.

R3C7: * Image Analysis and Immunofluorescence sections should contain a concise but detailed description and not require that readers look up a previous paper.

R3R7: We have added a description of all methods including dSTORM analysis and immunofluorescence.

Immunofluorescence "Cells were grown on coverslips coated with fibronectin. Between 1-2 days of confluency, samples were washed 1x with PBS, fixed, and washed 3x (5m) with PBS (0.5% Triton X-100, 0.5% Normal Goat Serum (NGS), and 0.5% Normal Horse Serum (NHS)). Samples were blocked for 30m at room temperature in blocking buffer (PBS with 0.5% Triton X-100, 5% NGS, 5% NHS, and 10 mg/mL bovine serum albumin (BSA)). Following blocking, samples were incubated at 37° C in a humidity chamber (1h and shaking at 35 rpm) with primary antibody. Samples requiring a secondary antibody were then washed 3x (5m) with PBS (0.5% Triton X-100, 0.5% Normal Goat Serum (NGS), and 0.5% Normal Horse Serum (NHS)) and incubated for 30m at 37° C with secondary antibody a concentration of 1:1000. Samples were then washed 3x (5m) with PBS and stored at 4° C protected from light until imaging."

dSTORM analysis "dSTORM images were exported at 4nm/pixel and analyzed using our in-house MATLAB (Mathworks, Natic, MA) analysis pipelines as described in Stahley et al., (2016) and Beggs et al., (2021). Desmosomes were manually selected and then automatically excised and aligned. Intensity was measured across the junction and averaged along the desmosome length. Linescans were normalized, smoothed and the plaque-to-plaque distance was quantified with the "peakfinder" function. See Fig. S1C for analysis details."

R3C8: * Statistical Analysis section should specify the individual tests that were applied.

R3R8: Thank you for this suggestion. We have added a new Statistical Analysis section to the Method section, reading as follows: *"We used the Shapiro-Wilks test to test for normality and the Brown-Forsythe test to test for equal variance of samples when 3 or more samples were assessed. If the Shapiro-Wilks and Brown-Forsythe tests requirements were met, One-way ANOVA with Tukey's multiple comparison tests were used. When the requirements of the Brown-Forsythe test*

were not met, a Brown-Forsythe and Welch ANOVA with Dunnett's t3 multiple comparison test was used. For 2 sample comparisons, the Shapiro-Wilks test to test for normality was used followed by an un-paired t-test."

Third decision letter

MS ID#: jcs.263906R2

MS Title: Desmoplakin tail domain position in the desmosomal plaque is isoform dependent

Authors: Collin M Ainslie; Krishna Patel; Yen T. B. Tran; Navaneetha Krishnan Bharathan; Samuel C. Bartley; Volker Spindler; Alexa L Mattheyses

Article Type: Short Report

Dear Dr Mattheyses,

I am happy to tell you that your manuscript has been accepted for publication in Journal of Cell Science, pending standard publication integrity checks.